# Disruption of the *cpsE* and *endA* Genes Attenuates *Streptococcus pneumoniae* Virulence: Towards the Development of a Live Attenuated Vaccine Candidate

**DOI:** 10.3390/vaccines8020187

**Published:** 2020-04-15

**Authors:** Malik Amonov, Nordin Simbak, Wan Mohd. Razin Wan Hassan, Salwani Ismail, Nor Iza A. Rahman, Stuart C. Clarke, Chew Chieng Yeo

**Affiliations:** 1Faculty of Medicine, Universiti Sultan Zainal Abidin, Kuala 20400, Malaysia; nordinsimbak@unisza.edu.my (N.S.); wmohdrazin@unisza.edu.my (W.M.R.W.H.); salwani@unisza.edu.my (S.I.); noriza@unisza.edu.my (N.I.A.R.); chewchieng@gmail.com (C.C.Y.); 2Faculty of Medicine and Institute for Life Sciences, University of Southampton, Southampton S016 6YD, UK; S.C.Clarke@soton.ac.uk; 3NIHR Southampton Biomedical Research Centre, University of Southampton, Southampton S016 6YD, UK; 4Global Health Research Institute, University of Southampton, Southampton S016 6YD, UK; 5International Medical University, Bukit Jalil, 57000 Kuala Lumpur, Malaysia

**Keywords:** *Streptococcus pneumoniae*, vaccine, live attenuated vaccine, capsule biosynthesis gene, *cpsE*, endonuclease A, *endA*

## Abstract

The majority of deaths due to *Streptococcus pneumoniae* infections are in developing countries. Although polysaccharide-based pneumococcal vaccines are available, newer types of vaccines are needed to increase vaccine affordability, particularly in developing countries, and to provide broader protection across all pneumococcal serotypes. To attenuate pneumococcal virulence with the aim of engineering candidate live attenuated vaccines (LAVs), we constructed knockouts in *S. pneumoniae* D39 of one of the capsular biosynthetic genes, *cpsE* that encodes glycosyltransferase, and the endonuclease gene, *endA*, that had been implicated in the uptake of DNA from the environment as well as bacterial escape from neutrophil-mediated killing. The *cpsE* gene knockout significantly lowered peak bacterial density, BALB/c mice nasopharyngeal (NP) colonisation but increased biofilm formation when compared to the wild-type D39 strain as well as the *endA* gene knockout mutant. All constructed mutant strains were able to induce significantly high serum and mucosal antibody response in BALB/c mice. However, the *cpsE-endA* double mutant strain, designated SPEC, was able to protect mice from high dose mucosal challenge of the D39 wild-type. Furthermore, SPEC showed 23-fold attenuation of virulence compared to the wild-type. Thus, the *cpsE*-*endA* double-mutant strain could be a promising candidate for further development of a LAV for *S. pneumoniae*.

## 1. Introduction

*Streptococcus pneumoniae* (pneumococcus) is responsible for about 1.2 million deaths globally, more than half of which happened in children aged <5 years [1]. Most pneumococcal-implicated deaths happened in developing countries such as India, Nigeria, the Democratic Republic of the Congo, and Pakistan, which constitute 49% of the global pneumococcal fatalities [2]. Mortalities are mainly due to pneumococcal pneumonia (81%) with the rest attributed to meningitis (12%), bacteraemia, and sepsis (7%) [2,3,4]. Even with modern antimicrobial treatment, survivors of invasive pneumococcal diseases usually develop long term complications such as hearing loss and neuropsychological impairments [5]. Still, these alarming figures of pneumococcal morbidity and mortality might underestimate the actual burden of pneumococcal diseases because many developing countries lack sensitive diagnostic tests for detecting the disease aetiology. 

Currently, two types of vaccines are used to prevent pneumococcal diseases: pneumococcal conjugate vaccines (PCV10 and PCV13) and pneumococcal polysaccharide vaccines (PPV23). The use of these vaccines has led to a considerable reduction in mortality ascribed to pneumococcal infections [2]. These vaccines are designed to generate antibodies against the capsular polysaccharide of the pneumococcus [6] which is structurally distinct among *S. pneumoniae* strains [7,8]. Currently used PCVs were initially designed for use in Western industrialized countries. Therefore, the vaccines would thus include the dominant serotypes causing invasive disease in those regions that are not well matched to other geographic areas of the world such as Asia and Africa. For example, in countries of the Asia–Pacific region, the potential serotype coverage of PCV13 ranges from 30% to 96.5% due to dissimilar serotype prevalence in different parts of the region [9]. Inclusion of new serotypes into the conjugate vaccine formula is a complex task and may further increase the price of the vaccine, thereby limiting its affordability for developing countries and would therefore not be a reasonable way of solving the problem. Even now, out of 144 countries which recently added PCV to their national immunisation programs, only 60 countries are categorized as low- or middle-income countries which could not afford PCV immunisation without support from the Gavi, the Vaccine Alliance [10,11]. 

Although vaccine serotypes have decreased after the introduction of the PCV’s, there has been an increase in non-vaccine serotypes in both carriage and invasive diseases due to serotype replacement [12,13] and clonal expansion [12,14,15,16,17]. Additionally, the polysaccharide nature of the vaccines, as compared to pure protein or peptide vaccines, does not provide efficient immunologic memory as T cell major histocompatibility complex (MHC) II receptors can only bind peptide antigens presented by B cell receptors [18]. Therefore, the immune response to polysaccharide vaccines is traditionally reported as T cell-independent and conjugation of polysaccharides to protein conjugates provides a T cell response [19]. Despite this, PCVs remain less efficient against more common non-invasive pneumococcal disease such as middle-ear infections, sinusitis, and bronchitis in comparison to invasive pneumococcal disease (IPD) [20,21]. 

The limitations outlined above necessitate the development of a serotype-independent vaccine that would confer protection in a serotype-independent manner against both mucosal and invasive pneumococcal disease, while keeping the cost sufficiently low to make it accessible for developing countries. *S. pneumoniae* live attenuated vaccines (LAVs) would ideally colonise in the upper respiratory tract of the host and induce effective immune protection by activating all phases of the immune system, provide more durable immunity that require boosters less frequently and have a lower cost. Attenuated vaccines are considered the most effective type of vaccine that induce durable immunity against diseases, with a lessened need for booster doses [22]. Unencapsulation is one of the effective strategies to attenuate the bacteria and to expose pneumococcal surface proteins to the immune system. In previous preclinical studies, mice immunised with unencapsulated strains of *S. pneumoniae* induced a robust immune response and protected the immunised animals against wild-type challenges [23,24]. We chose to knockout the *cpsE* gene as mutations in the gene has been associated with capsule loss, and increased pneumococcal colonisation, which is one of the predictors of effective immune response [25]. 

One of the biggest challenges in constructing LAVs is choosing a strategy that, besides attenuating virulence, would also eliminate the risk of reversion of the vaccine strain to a fully virulent strain because of the natural competence for transformation and the genetic plasticity of the pneumococcal genome. Therefore, besides unencapsulation by targeting one of the capsule synthesis genes (*cpsE* gene) we used a second knock out of the endonuclease *endA* gene that focuses on the safety of the candidate vaccine strains. Mutations in the *endA* gene in pneumococci have been reported to show decreased competence for natural transformation [26,27], thereby minimizing the potential for reversion of the constructed candidate vaccine strain into the wild-type. The goal of the current study was to determine the immune protection and attenuation of the constructed *endA* and *cpsE* gene knockout strains as a means of assessing their suitability as a candidate LAV.

## 2. Materials and Methods

### 2.1. Bacterial Strain and Growth Conditions 

*Streptococcus pneumoniae* was grown in Todd-Hewitt broth (THB; Oxoid, Basingstoke, Hampshire, UK) supplemented with 0.5% yeast extract (Oxoid, Basingstoke, Hampshire, UK), brain heart infusion (BHI) broth (Oxoid, Basingstoke, Hampshire, UK), or on blood agar (BA) plates (Isolab, Shah Alam, Selangor, Malaysia) at 37 °C and in 5% CO_2_. For storage, *S. pneumoniae* were grown in BHI broth at 37 °C to an OD_600_ of around 0.5, and 800 μL culture was frozen with the addition of 200 μL 80% glycerol (final concentration of glycerol in the stock media was 20%) and kept at −80 °C.

### 2.2. Construction of cpsE::tetL and endA::aphA-3 Knockout Mutants 

Knockout mutants of *cpsE* and *endA* genes in *S. pneumoniae* strain National Collection of Type Cultures (NCTC) 7466 (referred to as strain D39, serotype 2) were constructed by homologous recombination with disrupted gene modules as described previously [28,29,30]. Briefly, the disrupted gene modules were constructed by firstly amplifying the flanking regions (few hundred base-pairs) of the target genes which were then ligated to a selectable marker in the middle. The middle selectable marker used was either the *tetL* tetracycline resistance gene (to interrupt the *cpsE* gene) or the *aphA3* kanamycin resistance gene (to disrupt the *endA* gene) which were amplified from plasmid pMV158GFP [31] or from the genomic DNA of *S. pneumoniae* strain CP2460 (kindly gifted by Professor Don Morrison, University of Illinois, Chicago, IL, USA), respectively. In the second step, all the three PCR products were combined into one molecule by a splicing overlap extension-PCR (SOE-PCR) reaction. The resulting SOE-PCR product was validated by conventional Sanger dideoxy sequencing using appropriate primers prior to transformation into wild-type *S. pneumoniae* D39. Competence of *S. pneumoniae* D39 was controlled by the appropriate addition of competence-specific peptide [24] and transformation was performed as described previously [32]. Competence media used contained Todd-Hewitt broth supplemented with 0.2% yeast extract (THY), 0.01% CaCl_2_, 0.2% bovine serum albumin (BSA) and 100 ng/mL competence-stimulating peptide 1 (CSP-1; EMRLSKFFRDFILQRKK). Briefly, frozen pneumococcal competent cells (20 µL) were thawed on ice, diluted 1:10 in the competence medium (180 µL), and the cells were incubated at 37 °C for 15 min. Subsequently 10 µl of purified recombinant gene disruption modules (either *cpsE::tetL* or *endA::aphA-3* of approximately 30–50 ng) were added and the cells were incubated for 3 h at 37 °C before plating them on THY agar plates containing tetracycline (1.25 µg/mL) for the selection of *S. pneumoniae* D39 *cpsE::tetL* knockout mutants or kanamycin (200 μg/mL) to select *S. pneumoniae* D39 *endA::aphA-3* knockout mutants. Transformants obtained were validated by PCR and sequencing of the PCR products. To obtain the double knockout mutant, the D39 (*cpsE::tetL*) mutant, designated SPC, was made competent and was transformed with the *endA*::*aphA-3* module. The double knockout mutant was then selected on THY supplemented with both tetracycline and kanamycin and validated by PCR-sequencing. 

### 2.3. Transmission Electron Microscopy

Transmission electron microscopy was used to determine the presence of the capsule in the constructed mutant *S. pneumoniae* strains. Samples were prepared by lysine-acetate-based formaldehyde-glutaraldehyde ruthenium red-osmium fixation procedure (LRR fixation) as described previously [33]. The samples were examined with a transmission electron microscope (FeiTecnai G2 Spirit BIO TWIN, Hillsboro, OR, USA) at an acceleration voltage of 80 kV.

### 2.4. In Vitro Studies

Growth of the constructed mutant strains was assessed against wild type *S. pneumoniae* D39. The strains were grown in 5 mL BHI at 37 °C in 5% CO_2_ overnight in 15-mL tubes. From the overnight culture, 100 µL was transferred into a 15-mL tube containing 5 mL THY broth and grown to mid-log phase (OD_600_ = 0.5). Subsequently, 100 µL of the mid-log phase culture was transferred into a fresh sterile tube containing 5 mL THY and the optical density at 600 nm was measured every 30 min for 12 h using a spectrophotometer (Eppendorf BioSpectrometer plus, Hamburg, Germany). 

Biofilm formation analysis of the gene knockout strains was compared to the wild-type D39 strain using the Method described by Yadav, et al. [34]. Briefly, the constructed mutant strains and wild-type *S. pneumoniae* D39 were grown to a mid-logarithmic phase (OD_600_ = 0.4) and after 1:1000 dilution, 200 µL were transferred to a 96-well polystyrene microtiter plates. The plates were incubated at 37 °C in 5% CO_2_ and biofilm formation was assessed at 6, 12, 18, 24 and 36 h using the crystal violet assay as described elsewhere [34]. Optical density was measured at 570 nm using an Infinite M200 PRO Microplate Reader (Tecan, Männedorf, Switzerland). The biofilm assays were performed in triplicates and the means ± standard deviations were calculated. Background absorbance was compensated by subtracting the average values of crystal violet with the values obtained in sterile medium.

### 2.5. In Vivo Studies

Ethical approval (UAPREC/04/008) was obtained from the University Animal and Plant Research Ethics Committee to conduct experiments on animal models. The colonisation potential, immunogenicity, and protection against mucosal challenges of the constructed mutant *S. pneumoniae* strains were performed in BALB/c mice. For colonisation assay, six 10-12 week-old BALB/c mice in each group was inoculated with 10 μL phosphate-buffered saline (PBS) containing 10^7^ colony forming units (CFU)/mL of the relevant bacterial suspensions intranasally and at the 2nd, 10th, and 18th days post-inoculation, the mice were sacrificed, the nasopharyngeal (NP) fluid was collected, and bacteria were quantified by plating the NP fluid on BHI agar plates supplemented with 5% defibrinated blood and neomycin (20 μg/mL) to suppress the growth of contaminants. 

Mice immunisation was carried out by previously published methods (Rosch et al., 2007; Yasser 2013). The antibody titres against *S. pneumoniae* were determined by whole cell ELISA. Briefly, 5 groups of mice (each group consisting of 6 mice) was immunised with the constructed gene knockout strains, i.e., D39 (*cpsE::tetL*) (designated SPC), D39 (*endA::aphA3*) (designated SPE), and D39 (*cpsE::tetL*; *endA::aphA3*) (designated SPEC), and wild type D39 strain as a positive control to assess their ability to induce effective antibody responses. Mice were intranasally immunised by administering 10 µL PBS containing 1 × 10^7^ CFU of the bacteria in two doses, two weeks apart. Three weeks after the second dose, samples of blood and NP lavage fluid were collected for the measurement of antibody concentrations by enzyme-linked immunosorbent assay ELISA. Three-week periods were chosen because after 10 to 14 days following antigen exposure, IgG antibodies to protein antigens first appear in the blood and peaks within 4 to 6 weeks after primary immunisation [18]. Briefly, whole cell ELISA was conducted by fixing wild-type *S. pneumoniae* D39 in 96-well plates as antigens and blocking with 1% BSA. Subsequently, serum or nasal washes was added in 10-fold serial dilutions. Goat anti-Mouse IgG (Fc) and IgA (alpha chain) secondary antibodies were added and antibodies were measured with a 2,2’-azinobis (3-ethylbenzothiazoline-6-sulfonic acid)-diammonium salt (ABNS) substrate solution in citrate-phosphate buffer. Readings were carried out using an Infinite M200 PRO Multimode Microplate Reader at 493 nm and samples were considered negative for the presence of specific antibodies when OD_493_ < 0.1.

Protection against high dose intranasal challenge with the wild-type D39 strain was assessed after three weeks of immunisation. BALB/c mice were challenged with approximately 5 × 10^8^ CFU of wild-type *S. pneumoniae* D39 intranasally. Survival time (h) and rate of survival (%) was calculated using the Kaplan–Meier survival test.

The median lethal dose (LD_50_) in colony-forming units (CFU) was measured to evaluate the effect of the *endA* and *cpsE* mutation on the virulence of *S. pneumoniae* in BALB/c mice. Virulence of the strains was determined by the number of CFU required to kill 50% (LD_50_) of the exposed mice using the Reed and Muench formula [35]. 

Statistical analyses of in vitro growth potential, biofilm formation, in vivo colonisation density, and antibody levels between groups were assessed by analysis of variance (ANOVA) and post-hoc test. Comparison of survival between groups of mice was analysed with the Log Rank chi-squared test on the Kaplan–Meier survival data. In all cases, statistically significant differences were defined as *p* < 0.05. 

## 3. Results

### 3.1. Disruption of the cpsE Gene Led to Loss of the Capsule in S. Pneumoniae

To confirm that disruption of the *cpsE* gene by the *cpsE::tetL* knockout module affected the capsule of the recombinant pneumococcal mutant strain designated SPC, transmission electron microscope (TEM) was used to investigate the morphology of the pneumococcal cells (Figure 1). The parental *S. pneumoniae* D39 wild-type strain as well as the D39 (*endA::aphA3*) knockout strain, SPE, showed a thick layer of capsule whereas the capsule was absent in the *cpsE::tetL* knockout strains, SPC and SPEC.

### 3.2. In Vitro Growth Characteristics

The growth kinetics of the gene-disrupted mutant strains did not differ in the duration of the lag phase (around 2–3 h after inoculation in broth) from the D39 wild-type but the SPC and SPEC knockout strains showed much lower peak bacterial density (OD_600_ = 1.01 and 0.94, respectively) than the wild-type and SPE strain (OD_600_ = 1.49 and 1.38, respectively) in the transition from exponential growth to the stationary growth phases (Figure 2). There was a significant (*p* < 0.05) effect of the *cpsE* capsular gene knockout on the OD_600_ values starting from 6 h of growth. A Tukey post hoc test revealed that mutant strains containing the capsule gene knockout, *cpsE::tetL*, had significantly lowered peak bacterial density [SPC (OD_600_ = 0.923 ± 0.054; *p* < 0.01) and SPEC (OD_600_ = 1.032 ± 0.133; *p* < 0.01)] compared to the wild-type D39 strain (OD_600_ = 1.497 ± 0.005). However, no statistically significant difference was observed between the SPE strain and the wild-type strain.

### 3.3. In Vitro Biofilm Formation

All constructed gene-disrupted mutant strains and wild-type *S. pneumoniae* D39 were able to attach to the 96-well polystyrene surface; however, the *cpsE* gene knockout had significant effects on biofilm formation with post hoc comparisons using the Tukey test showing that the mean scores for the SPC and the SPEC strains were significantly different than the wild-type D39 strain at all the studied time intervals (Figure 3). However, for the SPE strain, biofilm formation did not significantly differ from the parental strain at any studied time intervals. The greatest biofilm formation capacity was observed at 18 h for the SPC (2.10 ± 0.11) and the SPEC mutant strains (2.09 ± 0.16), which were almost 20 times greater than the SPE (0.13 ± 0.01) and wild-type *S. pneumoniae* D39 (0.11 ± 0.01) strains. 

### 3.4. Density and Duration of Nasopharyngeal (NP) Colonisation in Mice

Since the longer colonisation duration of live attenuated vaccines in the NP mucosa induces better immune protection [36], the effect of the gene knockouts on NP colonisation of the pneumococcal mutants was determined. Following experimental colonisation by intranasal inoculation with 1 × 10^7^ CFU of each of the gene-disrupted mutant strains, *S. pneumoniae* colonies were recovered from the NP washes of the inoculated BALB/c mice at days 2, 10 and 18 post-inoculation (Figure 4) but had cleared by day 26 for all strains, including the D39 wild-type. Nevertheless, there were differences in the colonisation density and duration of mice for the different strains and at different time points. At 2 days post-inoculation, the highest colonisation density was observed in the D39 wild-type strain followed by the SPE, SPC and SPEC mutant strains. The highest colonisation density continued to be observed for the wild-type strain at days 10 and 18 post-inoculation. However, at day 18 of the colonisation study, half of the mice had cleared the bacteria in groups that were infected with the capsule mutant strains (i.e., SPC and SPEC) and the colonisation density was significantly lower when compared to the wild-type and SPE strains. The result suggests that the capsule gene knockout (i.e., *cpsE::tetL*) led to a decrease in the duration of NP colonisation in mice. At day 26, no bacteria were recovered from the NP wash fluids in all groups of mice. 

### 3.5. Effect of Immunisation with the Gene-Disrupted Mutant Strains on Local and Systemic Antibody Titres

To explore the antibody response of BALB/c mice to the constructed gene-disrupted pneumococcal mutant strains, levels of serum IgM, IgG, and IgA from NP wash samples were determined three weeks following two rounds of inoculation of 1 × 10^7^ CFU of the bacteria two weeks apart. Compared to the PBS-inoculated negative control group, levels of pneumococcal-specific IgG and IgA antibodies were high (*p* < 0.01) in all groups of inoculated mice; however, levels of IgM were significantly higher (*p <* 0.05) only in the SPEC-immunised group. Similar to serum IgG and NP mucosal IgA antibody levels, unencapsulation of the bacteria decreased serum IgM antibody response in the SPC strain. Compared to mock-immunised mice (2.40 ± 0.10) significantly higher IgM response was only observed with SPEC-immunised mice (2.57 ± 0.51; *p* < 0.05). Furthermore, the mice immunised with SPEC (2.57 ± 0.51; *p* < 0.01) and SPE (2.50 ± 0.43; *p* < 0.01) strains showed significantly higher IgM response compared to mice immunised with the SPC strain (2.26 ± 0.06) (Figure 5A). The mice immunised with the SPE knockout strain showed the highest level of IgG (1.51 ± 0.15; *p* < 0.01) compared to SPEC (0.99 ± 0.10), SPC (0.95 ± 0.12), and wild-type (1.02 ± 0.15) pneumococci-immunised mice (Figure 5B). However, the level of IgA antibodies was highest in mice that were immunised with wild-type pneumococci (1.17 ± 0.12; *p* < 0.01) compared to mice that were immunised with SPEC (0.76 ± 0.09), SPC (0.75 ± 0.10), and SPE (0.96 ± 0.08) mutant strains (Figure 5C). 

### 3.6. Survival Analysis

The ability of the constructed gene-disrupted mutant strains to protect against a lethal dose (≈5 × 10^8^ CFU) of intranasal wild-type *S. pneumoniae* D39 infection was assessed after immunisation of mice with each of the constructed mutant strains. Kaplan-Meier survival analysis showed that mice immunised with the SPEC strain demonstrated the highest survival rate (56%) and significantly longer survival time 176.0 h (95% CI 136.2–228.6) compared to the SPC strain immunised 99.0 h (*p* < 0.05, 95% CI 52.6–145.4) and mock-immunised mice 96.0 h (*p* < 0.05, 95% CI 57.5–134.5) (Figure 6). Mice immunised with the wild-type strain showed 50% survival rate and 172.2 h (95% CI 123.7–220.7) survival time was also significantly different compared to survival time of mice immunised with SPC and mock-immunised mice. Moderate survival rate (40%) and time 148.8 h (95% CI 98.7–198.9) were observed in mice immunised with the SPE strain; however, the survival time was not significantly different compared to the other groups. In contrast, mice immunised with the SPC strain showed the lowest survival rate and time, 20% and 99.0 h (±23.7), respectively. 

### 3.7. Attenuation of Virulence

Virulence of the constructed gene-knockout strains was measured as the lethal dose for BALB/c mice. The median lethal dose (LD_50_) in CFUs was measured by intranasal infection of four groups of ten mice each with 5-fold serial dilutions. Since the highest protection was observed in mice immunised with the SPEC mutant strain (see “Survival analysis” subsection above), we studied the level of attenuation of this strain as compared to the wild-type *S. pneumoniae* D39.

Although both parental and SPEC strains were grown to OD_600_ = 0.8, the SPEC strain showed almost 10 times higher CFU/mL, i.e., 1.46 × 10^10^ as compared to 1.42 × 10^11^ for the wild-type D39 strain, at which concentrations both strains showed 100% lethality in mice. The LD_50_ for the intranasal parental wild-type D39 strain was 1.5 × 10^9^ CFU/mL. However, the SPEC double-mutant strain showed 23 times higher LD_50_ (at 3.5 × 10^10^ CFU/mL) compared to the parental wild-type strain. Thus, the SPEC double-knockout mutant strain displayed a 23-fold level of attenuation of virulence when compared to the wild-type D39 strain (Table 1).

## 4. Discussion

Unencapsulation is one of the strategies to unmask subcapsular protein antigens [38] in order to induce immune response against multi-epitope subcapsular proteins [39]. An anti-protein immune response is crucial for anti-pneumococcal immunity as non-encapsulated strains are emerging following the widespread use of capsule-based vaccines (i.e., PCV and PPV) worldwide [40]. 

The polysaccharide capsule in *S. pneumoniae* is synthesised from a single capsule operon (designated *cps*) which consists of 17 genes in the D39 strain [41]. Deletions or single base-pair mutations in any of the capsule operon genes affect expression of the polysaccharide capsule and pneumococcal virulence. For example, deletion of *cpsA* causes partial encapsulation of *S. pneumoniae*, and deletion of *cpsB* and *cpsD* or point mutations in *cpsE* gene led to the deletion of the entire pneumococcal capsule [25,42]. 

The *cpsE* gene is responsible for the addition of activated sugars to the lipid carrier in the bacterial membrane [43,44] and, in this study, a *cpsE* knockout led to the unencapsulation of the resulting recombinant *S. pneumoniae* mutant strain. The *cpsE* gene has been reported to be essential for *S. pneumoniae* encapsulation in serotypes 13, 14, 15B, and 19F [45,46]. A single mutation in the *cpsE* gene was found to cause loss of capsule expression [25]. Previously, unencapsulated pneumococcal live vaccines were constructed by total capsule operon deletion [24]. We avoided the entire operon deletion because the capsule synthesis genes are also involved in other pneumococcal metabolic processes besides polysaccharide capsule production [47]. Even some unencapsulated pneumococcal human isolates were found to preserve the capsule operon [48]. However, we cannot exclude the possibility that insertion of the antibiotic resistance marker into the *cpsE* gene could have a polar effect in the expression of downstream genes. 

### 4.1. Capsule Mutations Affected the Growth of S. Pneumoniae

To induce vaccine-mediated protective response, live vaccines should persist within the host in an appropriate location in sufficient quantities and for a substantial duration of time and should contain both antigen and immune-activating endogenous danger signals as adjuvants [49,50].

The in vitro growth behaviour of constructed knockout strains could predict certain characteristics of *S. pneumoniae*, especially the invasiveness and colonisation potential of the pneumococcal strains. Any alteration in the capsular operon genes (even single point mutations) could cause changes from decreased levels of capsule production to the complete loss of the capsule [51,52] and may alter growth, adherence, invasiveness, colonisation, and competence of the microorganism [25,53].

This study showed that the *cpsE* gene knockout led to a complete loss of the polysaccharide capsule in the SPC and SPEC mutant strains (Figure 1). The capsule knockout strains did not, however, show any differences in the initial lag phase of the growth compared to the D39 wild-type parental strain. In contrast, an earlier study had showed that deletion of several downstream capsule genes such as *cpsK*, *cpsP*, *cpsK–O*, and *cpsM–O* led to a significantly prolonged lag phase when compared to wild-type *S. pneumoniae*. However, deletion of other serotype-specific capsule genes such as *cpsF–O* and *cpsL–O* or several other genes which are downstream of *cpsK* did not significantly changed the duration of the in vitro lag phase [54]. A possible reason for the slowing of the lag phase in these mutants is that the proteins and enzymes encoded by the capsule genes are also involved in other bacterial metabolic pathways besides polysaccharide capsule production. Therefore, during its growth the mutant bacteria may need time to switch to alternative pathways leading to the observed longer lag phase [54]. Schaffner et al. (2014), who worked on clinical *S. pneumonia* serotype 18C isolate, also demonstrated an increase in the lag phase in naturally-occurring unencapsulated *S. pneumoniae* strains along with a laboratory-constructed unencapsulated strain by point mutations in the *cpsE* gene [25]. 

We found that the *cpsE* mutant strains had significantly lower maximal OD_600nm_ value (i.e., at stationary phase) compared to that of wild-type and the *endA*-disrupted mutant strain, SPE. After the exponential (log) phase, the bacterial growth slowed down most likely from a shortage of nutrients, thus entering the stationary phase. However, encapsulated bacteria could utilize their capsular carbohydrates for their nutritional needs. This agrees with the findings of Hamaguchi et al. (2018) who demonstrated clear survival advantage of capsulated strains of *S. pneumoniae* in aqueous media devoid of nutrients compared to unencapsulated strains [55]. Nonetheless, this finding appeared to be serotype-specific, as Schaffner et al. (2014) who studied point mutation in naturally occurring *cpsE* (C to G change at gene position 1135) which rendered pneumococci non-encapsulated showed opposite data where mutant *S. pneumonia* had the maximal OD_600nm_ values compared with encapsulated strain of serotype 18C [25]. 

### 4.2. Biofilm Profiles of the Candidate Vaccine Strains

Growth in biofilm enables *S. pneumoniae* to not only resist the bactericidal effects of antimicrobial agents and to help in evading the host immune system, but to also facilitate microbial persistence and their dissemination [56]. Studies of pneumococcal gene expression in in vitro biofilm models showed altered production of several virulence factors including capsular polysaccharides [57,58,59]. 

Although previous reports have shown altered biofilm formation when the capsule operon, or specific genes of the operon were deleted, the biofilm formation of a specific *cpsE* gene-knockout strain has not been investigated. We found that the *cpsE* mutation significantly increased in vitro biofilm formation by several folds in *S. pneumoniae*. This is likely due to exposure of sub-capsular adherence factors to the surface of polystyrene plates as a result of impaired polysaccharide capsule in these mutants. This finding broadly supports the work of others who have reported that impaired pneumococcal capsule led to increased biofilm formation. Qin et al. (2013) demonstrated unencapsulation by knocking out the *cpsD* gene which resulted in an increase in biofilm formation but decreased virulence in a *S. pneumoniae* TIGR4 strain. In vivo studies also showed that pneumococcal strains that lack polysaccharide capsules form much stronger biofilms than encapsulated ones [60]. 

We hypothesised that the *endA* gene might also be involved in pneumococcal biofilm formation since a recent study showed that *endA* gene knockouts led to the impairment of biofilm formation in *Pseudomonas aeruginosa* [61]. However, our data showed that in *S. pneumoniae*, biofilm formation is not affected by *endA* interruption since there was no difference in biofilm density between the wild-type parental strain and the *endA* knockout strain. The pneumococcal-encoded *endA* may thus serve a different function in relation to biofilms as compared to the *P. aeruginosa*-encoded *endA*. 

### 4.3. Colonisation Potential of the Pneumococcal Gene-Knockout Strains

Pneumococcal polysaccharide capsule is an important virulence factor required for both colonisation and disease severity [62,63]. In a mouse model, we demonstrated that unencapsulated mutant strains of *S. pneumoniae* D39 were capable of NP colonisation; however, compared to the wild-type parental strain, unencapsulated strains had a reduced duration and density of colonisation. It can be hypothesised that partially unencapsulated *S. pneumoniae* strains or strains that were completely devoid of the polysaccharide capsule are more susceptible to phagocytosis by polymorphonuclear leukocytes (PMNs) [64]. Nevertheless, the variability of the colonisation potential among unencapsulated strains of different capsular serotypes inferred that the colonisation potential appeared to be serotype-specific. Previously constructed unencapsulated mutants of serotypes 6A and 4 by capsular operon deletion reported significantly decreased colonisation density and duration when compared to their respective wild-type strains; however, serotype 4 had higher colonisation density than serotype 6A. Furthermore, the study did not find any association of colonisation density with protective immune responses [24]. Findings of our study also revealed that the constructed *endA* and *cpsE* mutant strains and the wild-type parental D39 strain sustained longer colonisation duration compared to that of other previous reports [24,65]. Pneumococcal colonisation of the upper NP of mice remained for more than 18 days for all strains but in half of the group of mice that were administered with the unencapsulated SPEC and SPC mutant strains, the bacteria were non-detectable and presumably cleared by the host after 18 days. The unencapsulated strains also had lower cell density compared to the capsulated strains. An earlier study conducted with the *S. pneumoniae* TIGR4 strain showed colonisation duration of only 14 days for unencapsulated mutants whereas encapsulated strains continued to be recovered from NP washes after that period [65]. Variability in host colonisation duration in different vaccine studies could be explained by the use of different pneumococcal serotypes and host-related factors such as the strain of mice used in these studies [66].

The results of this study also indicated that the colonisation potential of *S. pneumoniae* was not affected by deletion of the *endA* gene. We initially postulated that the *endA* knockout might also affect colonisation duration because of its reported role in mediating resistance against neutrophil-mediated killing [67,68]. However, our data suggested that *endA* mutation in *S. pneumoniae* strain D39 did not lead to any loss in its capability to colonise the upper respiratory tract of mice. Perhaps the role of *endA* in resistance towards neutrophil-mediated killing needs to be reassessed in the D39 strain and could be another serotype-dependent characteristic of *S. pneumoniae*. 

### 4.4. Antibody Response to Candidate Vaccine Strains

In the serum, the majority of natural antibodies are composed of IgM which are reactive with many conserved pneumococcal epitopes. These antibodies limit the progression of infections during the early stages of inflammation and contribute, through somatic hypermutation and affinity maturation process, to produce high affinity specific IgG and IgA antibodies [69]. In the secondary immune response lower affinity adaptive IgM antibodies are also produced by IgM+ memory B cells which have enhanced complement activation property that protect the organism from pathogenic infection [70]. Our study also demonstrated that high level of pneumococcus specific IgM antibodies correlate with better protection in SPEC immunised mice. Other groups of mice immunised with single knockout strains, including wild type, did not show significantly higher IgM compared to mock immunised mice which is in agreement with Jason et al. (2014) who compared the IgM levels of mice immunised with a *ftsY*-knockout pneumococcal strain [28]. Furthermore, mice immunised with *cpsD-*, *lgt-*, and *pabB*-knockout D39 pneumococcal strains demonstrated better survival than mock immunised mice although those strains did not show significantly different IgM levels [36]. 

The levels of pneumococcus-specific IgA in NP lavage and IgG in serum were significantly increased after immunisation with the gene-disrupted strains SPEC, SPE, SPC as well as the wild-type parental D39 strain as compared to mock-immunised mice. We initially postulated that unencapsulation will increase the antibody response against subcapsular proteins since the polysaccharide capsule hides or masks these proteins from the host antibody recognition. However, there was no difference in serum IgG levels between the unencapsulated mutant strains (i.e., SPC and SPEC) and the encapsulated parental wild-type *S. pneumoniae* D39 strain. Roche et al., (2007) also demonstrated high serum IgG levels after immunisation with an unencapsulated strain and capsule-intact *ply*/*pspA-*knockout 6A strains [24]. Similar to our findings, there was no significant difference in the IgG levels between these groups of mice. Furthermore, experiments with µMT mice (a mice strain that lack mature B cells to produce specific antibodies) and MHCII^−/−^ mice that lack CD4*^+^* T cells demonstrated absence of protection following immunisation indicating the importance of antibodies and CD4^+^ T cells in mucosal and systemic protection [24]. 

In our study, the serum IgG and NP lavage IgA levels of mice immunised with the SPC, SPE, and SPEC strains and the wild-type parental D39 strain were greater than mock immunised mice. However, the serum IgG levels of mice immunised with the SPE strain were significantly higher than that of mice immunised with the SPC, SPEC, and wild-type D39 strains. The reason for the higher serum IgG response to the SPE strain is possibly due to the increased sensitivity of the strain to neutrophil-mediated phagocytosis and neutrophil activation of the adaptive immune system by engaging with lymphocytes and antigen-presenting cells [71]. Although immune protection against pneumococcal infections after NP colonisation was found to be mostly dependent on antibodies that recognise several *S. pneumonia* cell wall proteins in mice, in humans the dominant role of anti-protein antibodies over anti-capsule antibodies still remains unclear. In recent years, however, the growing number of evidence showed the role of serum anti-protein antibodies (IgG) to protect from systemic invasive infection and Th17 CD4^+^ cells for mucosal infections [72,73,74,75]. Furthermore, prospective cohort studies in humans also prove the role of antibodies and T cells in the protection from pneumococcal infections [76,77]. 

To achieve protection against *S. pneumoniae* infection, it is necessary to prevent pneumococcal NP carriage since invasion of the microorganism is usually preceded by at least a short duration of carriage. Immunity against pneumococcal carriage is provided by passive mucosal IgA antibody and L-17-dependent T cell immunity [78]. In this study, all the constructed gene-knockout strains (i.e., SPC, SPE, SPEC) and wild-type parental D39 strain induced significantly high IgA antibody levels in mice NP lavage. The IgA levels of mice induced by the wild-type strain were higher than the IgA levels induced by the gene-knockout strains. Similar findings were obtained by Ibrahim et al. with the pneumococcal HtrA mutant strain which is sensitive to high temperatures and oxidative stress [79,80]. However, Cohen et al. (2012), showed that the IgA response to their candidate vaccine strains (a D39Δ*lgt* and a D39Δ*pab* strains) was similar to the wild-type parental strain. 

### 4.5. Immune Protection Conferred by Gene-Knockout Strains

In this study, all the gene-knockout strains conferred better protection against very high doses of wild-type mucosal challenge compared to mock (PBS)-immunised mice. The highest protection was observed in mice immunised with the double mutant strain (SPEC) followed by the wild-type parental strain. Even though mice immunised with the SPE strain demonstrated the highest serum IgG level, protection from mucosal challenge was lower than mice immunised with SPEC and wild-type parental strain. This discrepancy between pneumococcus-specific serum IgG level and the protection against lethal challenges can perhaps be explained by the in vitro nature of antibody measurement used in this study. Although serum IgG from immunised mice bind to pneumococci whole-cells during ELISA, they might not be functional during in vivo protection. Cohen et al. (2012) obtained similar findings with high serum IgG levels but lower protection from heterologous challenge. They postulated that the discrepancy might be due to bacterial lysis that occurred during overnight incubation which enabled intracellular antigens to become accessible to serum antibodies in whole cell ELISA [81]. 

The lower level of protection observed in this study may also be due to the use of the highly encapsulated pneumococcal strain, D39, in the lethal challenge as it has been shown that capsule thickness affects opsonophagocytosis, which reduces as the thickness of the capsule increases [82]. However, in our research the wild-type strain conferred excellent protection in mice. In our experiments, we used a higher challenge dose (5 × 10^8^ CFU) of the wild-type strain compared to that used in the previous studies (1 × 10^7^ to 5 × 10^7^ CFU) [24,28,36]; this could be one of the reasons why we did not achieve 100% protection. On the other hand, the genetic background of the animal models used should also be considered in studying pneumococcal immunity. Although BALB/c mice, which was used in this work, is preferred for immunity studies, it is fairly resistant to *S. pneumonia* lethality compared to CD1 and C57BL/6 mice [83,84]. 

Since intranasal challenge studies showed that SPEC and the wild-type parental strain conferred the highest protection in mice, we further investigated the attenuation of virulence of these strains by measuring the LD_50_ dose, i.e., the dose which causes the death of 50*%* of mice. It was found that the double-knockout SPEC strain showed 23-fold of attenuation in the BALB/c mice when compared to the wild-type parental D39 strain. It is likely that the level of attenuation was due to unencapsulation as a result of the *cpsE* gene knockout as the polysaccharide capsule is considered one of the most important virulence factors of *S. pneumoniae* [65].

Here, we also did not directly investigate the effect of the *endA* gene knockout on virulence, but we postulated that the *endA* knockout would also have a considerable contribution in the attenuation of pneumococcal virulence observed in the SPEC mutant. We were unable to measure the level of attenuation for the single knockout strains as this would have been prohibitive in terms of cost as well as infrastructure due to the large number of mice involved in assessing the LD_50_ dose. Previous studies have shown that pneumococcal infections caused abundant neutrophil infiltration which, upon activation, releases neutrophil extracellular traps (NETs), which bind and kill pneumococci. However, pneumococci escape these neutrophil NETs by degrading the DNA component of NETs using a cell-surface endonuclease encoded by *endA* [85,86]. Therefore, it was postulated that exposure of the host to the *endA* knockout strain and the resultant increased neutrophil killing would also lead to attenuation of the SPEC strain. It was previously reported that vaccination with a *pep27* mutant strain of *S. pneumoniae* had led to increased neutrophil killing of the bacteria, and this, in turn, caused the more than 50-fold attenuation in intranasal administration [37]. Unfortunately, most pneumococcal live vaccine studies did not evaluate the fold of attenuation of the constructed mutant strains [28,29,79]. 

## 5. Conclusions

Our findings clearly demonstrated that the *cpsE* capsule gene knockout decreased peak bacterial growth, colonisation density and duration and furthermore increased biofilm formation in the constructed *S. pneumoniae* SPE and SPEC gene knockout strains. However, no differences were observed in these experiments between the wild-type and SPE strains which indicated that the *endA* gene likely had no observable role in biofilm formation and growth of *S. pneumoniae.* Our findings provided additional evidence that unencapsulation decreased serum IgG antibody levels. Although mice immunised with the SPE strain showed highest anti-pneumoccal IgG levels, we did not find better survival advantage in mice immunised with the SPE strain compared to mice immunised with other candidate vaccine strains. Mice immunised with the SPEC strain showed the highest survival rate and longest survival time after a lethal dose challenge when compared to the single gene knockout strains. Furthermore, the virulence of the SPEC double-mutant was attenuated by 23-fold compared to the parental wild-type strain. Nevertheless, the process of attenuation might also decrease the immunogenicity of candidate LAVs, because attenuation is mostly done by knocking out the pathogenic mechanisms of the microorganism. Those pathogenic mechanisms are usually responsible for inducing robust immune responses as well. In this instance, no drastic decrease in the immune response was observed in mice that were immunised with the SPEC double-mutant strain. Thus, the SPEC double-mutant strain could be considered a promising candidate for a LAV strain. Further studies to compare its immune response with heat-inactivated *S. pneumoniae* and currently available vaccines will be conducted in the future. 

## Figures and Tables

**Figure 1 vaccines-08-00187-f001:**
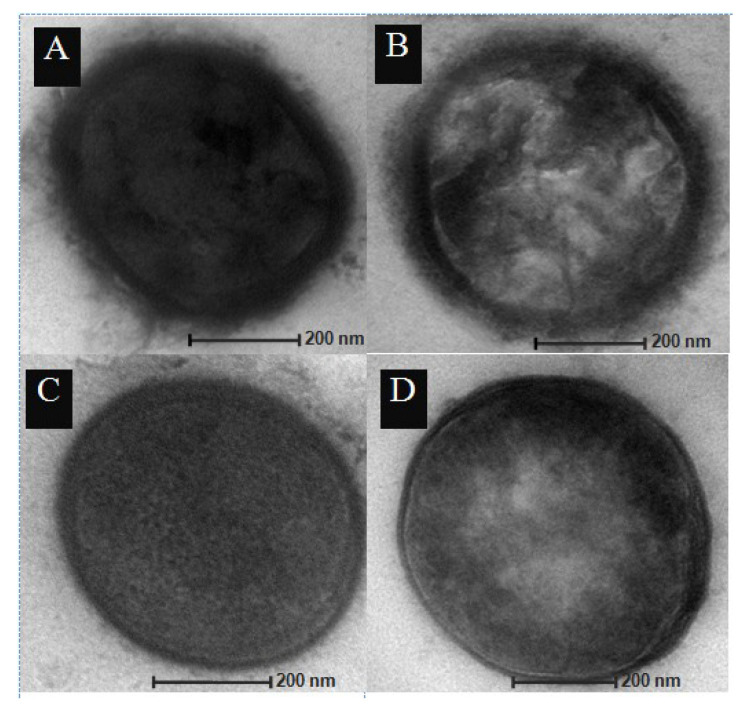
Detection of capsule by transmission electron microscope (TEM). (**A**) wild-type parental D39 strain; (**B**) SPE strain; (**C**) SPC strain; (**D**) SPEC strain.

**Figure 2 vaccines-08-00187-f002:**
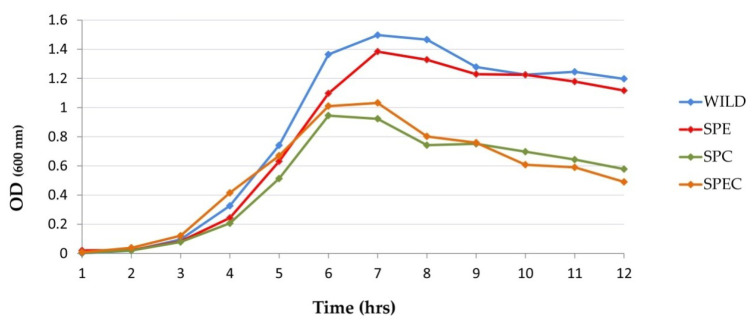
In vitro growth curve of *S. pneumoniae* constructed mutant strains as compared to the wild-type D39 strain in brain heart infusion (BHI) media as determined by the OD_600_ values with time of inoculation. The statistical analysis of growth potential between the groups was performed using analysis of variance (ANOVA) followed by a post hoc Tukey’s test from three independent experiments. Abbreviations used: wild, *S. pneumoniae* D39 wild-type strain; SPE, *S. pneumoniae* D39 (*endA::aphA3*) knockout strain; SPC, *S. pneumoniae* D39 (*cpsE::tetL*) knockout strain; SPEC, *S. pneumoniae* D39 (*cpsE::tetL*; *endA::aphA3*) double-knockout strain.

**Figure 3 vaccines-08-00187-f003:**
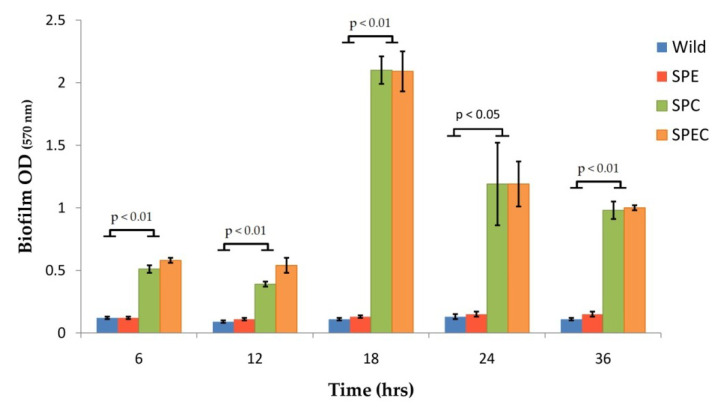
Biofilm formation capacities of *S. pneumoniae* D39 wild-type, SPE, SPC, and SPEC gene-disrupted mutantstrains in a 96-well microtiter plate as measured by the OD_570_ values following crystal violet staining. Biofilm formation between the groups was assessed by analysis of variance (ANOVA) and post-hoc test. Abbreviations used: wild, *S. pneumoniae* D39 wild-type strain; SPE, *S. pneumoniae* D39 (*endA::aphA3*) knockout strain; SPC, *S. pneumoniae* D39 (*cpsE::tetL*) knockout strain; SPEC, *S. pneumoniae* D39 (*cpsE::tetL*; *endA::aphA3*) double-knockout strain.

**Figure 4 vaccines-08-00187-f004:**
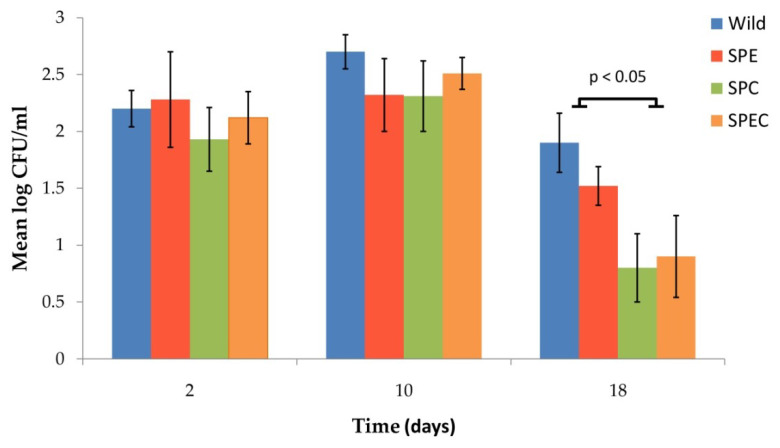
Number of *S. pneumoniae* cells (presented as log_10_CFU/mL) recovered from the nasopharyngeal wash fluids of BALB/c mice at various time intervals following intranasal inoculation with 1 × 10^7^ CFU of the respective bacterial strains. Error bars represent standard deviation (SD). The statistical analysis of nasopharyngeal (NP) colonisation between the groups was performed using analysis of variance (ANOVA) followed by a post-hoc Tukey’s test. Abbreviations used: wild, *S. pneumoniae* D39 wild-type strain; SPE, *S. pneumoniae* D39 (*endA::aphA3*) knockout strain; SPC, *S. pneumoniae* D39 (*cpsE::tetL*) knockout strain; SPEC, *S. pneumoniae* D39 (*cpsE::tetL*; *endA::aphA3*) double-knockout strain.

**Figure 5 vaccines-08-00187-f005:**
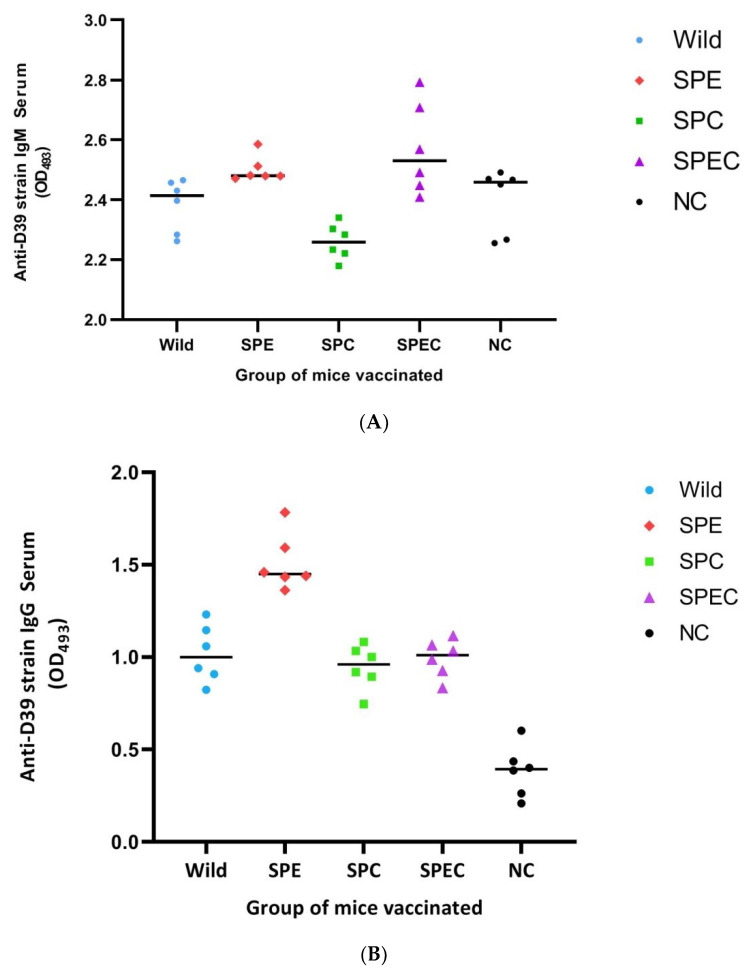
Comparison of the anti-D39 strain serum IgM (**A**), IgG (**B**), and NP lavage IgA (**C**) antibody levels of mice after vaccination with the indicated strains. The serum utilised was diluted to 1:100 and NP fluid was diluted to 1:10. Bars represent the respective mean values. The statistical analysis of antibody levels between the groups was performed using analysis of variance (ANOVA) followed by a post hoc Tukey’s test. In **(A)**, significant differences were found between groups SPEC and NC (*p* < 0.05), between groups SPEC and SPC (*p* < 0.01), and between groups SPE and SPC (*p* < 0.05). In **(B)**, significant differences (*p* < 0.01) were observed in all groups except between Wild-, SPC- and SPEC-immunised mice. In **(C)**, significant difference (*p* < 0.01) were found in all groups except between SPC- and SPEC-immunised mice. Abbreviations used: Wild, *S. pneumoniae* D39 wild-type strain; SPE, *S. pneumoniae* D39 (*endA::aphA3*) knockout strain; SPC, *S. pneumoniae* D39 (*cpsE::tetL*) knockout strain; SPEC, *S. pneumoniae* D39 (*cpsE::tetL*; *endA::aphA3*) double-knockout strain; and NC, negative control, i.e., mice vaccinated with phosphate-buffered saline (PBS).

**Figure 6 vaccines-08-00187-f006:**
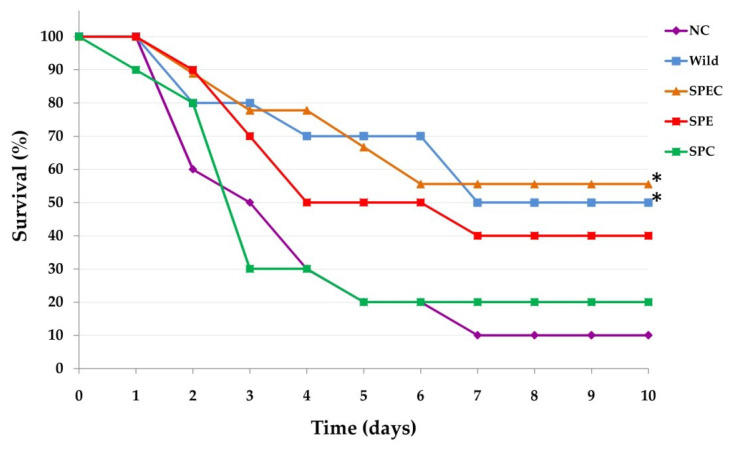
Survival rate (%) of vaccinated mice after exposure to a lethal dose (≈5 × 10^8^ CFU) of intranasal wild-type *S. pneumoniae* D39 infection. Comparison of survival between groups of mice was analysed with the Log Rank chi-squared test on the Kaplan–Meier survival data. * pairwise comparisons show SPEC and wild strain immunised mice had significant differences (*p* < 0.05) between SPC and mock immunised mice. Abbreviations used: NC, mice immunised with PBS; Wild, mice immunised with wild-type *S. pneumoniae* D39; SPE, mice immunised with *S. pneumoniae* D39 (*endA::aphA3*) knockout mutant; SPC, mice immunised with *S. pneumoniae* D39 (*cpsE::tetL*) knockout mutant; SPEC, mice immunised with *S. pneumoniae* D39 (*cpsE::tetL*; *endA::aphA3*) double-knockout mutant.

**Table 1 vaccines-08-00187-t001:** Attenuated virulence of live SPEC strain in vivo.

Strains	LD_50_ (CFU)	Fold of Attenuation *
SPEC	3.5 × 10^10^	23
Wild-type D39	1.5 × 10^9^	1

* Fold attenuation is calculated by dividing LD_50_ of the SPEC mutant strain to LD_50_ of the wild-type [37].

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
