# Peer review of "Disruption of the cpsE and endA Genes Attenuates Streptococcus pneumoniae Virulence: Towards the Development of a Live Attenuated Vaccine Candidate"

_vaccines, 2020, doi:10.3390/vaccines8020187_

Round 1

Reviewer 1 Report

In the manucript by Amonov et al., the authors generated S. pneumonia live attenuated vaccine candidate by disrupting the cpsE and endA genes, and evaluated the mutant strain for its attenuation, immunogenicity, and protective ability against S. pneumonia. The concept to develop a live attenuated vaccine for S. pneumonia is interesting to the field. However, the study has a major weakness that it lacks appropriate controls, such as heat-killed D39 strain and PCV. In addition, the authors compared the mutant strains to the wild-type strain on testing the protective efficacy against pathogen, and found that immunization with the mutant strains was no better than the wild-type strain. This observation is puzzling and does not support the conclusion that the mutant strain could be a promising candidate. Furthermore, assessing median lethal dose is not sufficient to prove the safety of the live attenuated vaccine.

Specific comments:

  1. The study lack important controls. For example, heat-killed D39 strain and PCV should be included in immunization experiments that assess host immune responses induced by the vaccine and protective efficacy of the vaccine.
  2. In the methods, the antibody titer was measured by serially diluting of serum, but the Figure 5 shows the end-point value. The dilution factor should be mentioned. In addition, it would be interesting to know IgM reactivity.
  3. Figure 6 lacks statistic analysis.
  4. To prove attenuation of virulence, assessing bacteria burden in organ after immunization is more appropriate than estimating LD50 alone.

Reviewer 2 Report

The aim of the manuscript was to attenuate pneumococcal virulence with the aim of engineering candidate live attenuated vaccines (LAVs). The authors constructed knockouts in S. pneumoniae D39 of one of the capsular biosynthetic genes, cpsE, and the endonuclease gene, endA, that had been implicated in the uptake of DNA from the environment as well as bacterial escape from neutrophil-mediated killing. The conclusion is that  cpsE-endA double-mutant strain could be a promising candidate for further development of a LAV for S. pneumoniae

  1. It is an innovative study in a relevant topic.
  2. The authors provide relevant research data and conclusions.
  3. The manuscript is well-written and is easy to understand. The Abstract is concise and summarizes the article content.
  4. The Introduction represents an adequate synthesis of the literature.
  5. It is suggested to add the aim of the study to the end of Introduction removing this sentence from the conclusion (lines 515-516).

Round 2

Reviewer 1 Report

The authors have addressed the comments well, and revised their manuscript accordingly. Though there are some limitations in the current study, the authors have discussed such limitations and proposed for the future study. I have no more comments.